# Derivation of Human Extraembryonic Mesoderm-like Cells from Primitive Endoderm

**DOI:** 10.3390/ijms241411366

**Published:** 2023-07-12

**Authors:** Karin Farkas, Elisabetta Ferretti

**Affiliations:** 1Novo Nordisk Foundation Center for Stem Cell Biology (DanStem), University of Copenhagen, 1165 Copenhagen, Denmark; karin.farkas@sund.ku.dk; 2Department of Biomedical Sciences, University of Copenhagen, 2200 Copenhagen, Denmark; 3Department of Cellular and Molecular Medicine, University of Copenhagen, 2200 Copenhagen, Denmark

**Keywords:** extraembryonic mesoderm, primitive endoderm, hypoblast, yolk sac, vasculogenesis, placenta, human gastrulation

## Abstract

In vitro modeling of human peri-gastrulation development is a valuable tool for understanding embryogenetic mechanisms. The extraembryonic mesoderm (ExM) is crucial in supporting embryonic development by forming tissues such as the yolk sac, allantois, and chorionic villi. However, the origin of human ExM remains only partially understood. While evidence suggests a primitive endoderm (PrE) origin based on morphological findings, current in vitro models use epiblast-like cells. To address this gap, we developed a protocol to generate ExM-like cells from PrE-like cell line called naïve extraembryonic endoderm (nEnd). We identified the ExM-like cells by specific markers (*LUM* and *ANXA1*). Moreover, these in vitro-produced ExM cells displayed angiogenic potential on a soft matrix, mirroring their physiological role in vasculogenesis. By integrating single-cell RNA sequencing (scRNAseq) data, we found that the ExM-like cells clustered with the *LUM*/*ANXA1*-rich cell populations of the gastrulating embryo, indicating similarity between in vitro and ex utero cell populations. This study confirms the derivation of ExM from PrE and establishes a cell culture system that can be utilized to investigate ExM during human peri-gastrulation development, both in monolayer cultures and more complex models.

## 1. Introduction

Modeling early human embryogenesis is essential to advance research in infertility treatment, pregnancy complications, and prevention of congenital diseases. It has been reported that 50–70% of conceptions result in early pregnancy loss, which is the most common reproductive complication that is unrecognized in most cases [1]. Early embryonic development involves extensive cellular rearrangements, including implantation, gastrulation, and establishment of the placental circulation required for the maternal–fetal exchange [2,3]. ExM is an understudied supportive structure that emerges early in development around gastrulation [4]. It forms the yolk sac, which is the site of primitive hematopoiesis, giving rise to erythroid and myeloid progenitors, and the source of endothelial progenitors forming vasculature [4,5,6]. Human ExM also contributes to chorionic villi in the placenta that infiltrate the endometrium and connect to the maternal circulation [7,8], and the allantois that links the embryo to the placenta [9]. Thus, studying ExM formation, development, and interaction with the surrounding tissues could shed new light on the mechanisms of infertility and pregnancy complications.

While mouse peri-gastrulation embryogenesis has been comprehensively characterized in vivo and in vitro [10,11], human development at this stage remains enigmatic, due, part, to the inaccessibility of human embryos and ethical constraints [12,13]. Recently, innovative approaches have challenged these barriers [14,15]. Novel methods were developed to study human in vitro-grown embryos until the post-implantation stage, just before the onset of the primitive streak (PS) [16,17]. A combination of single-cell sequencing and bioinformatics tools has enabled lineage tracing using DNA mutations [18] and gene expression analysis in the gastrulating embryo at the single-cell resolution [19], enabling the definition of cell-type specific markers. Lately, the first gastrulation model was generated in micro-patterned 2D cultures [20]. Three-dimensional models, including blastoids, gastruloids, and assembloids, incorporating multiple cell types, were introduced around the same time, allowing the study of complex processes such as symmetry breaking and/or implantation [15,21,22,23,24,25].

To examine human peri-gastrulation embryos and to generate suitable models by assembling multiple cell types, culturing conditions for the respective cell lineages must be established. To this end, ExM represents a critical lineage, but due to the lack of data from early human development, the origin of human ExM has not been fully established [4]. While studies in the mouse suggest that murine ExM is derived from the PS during gastrulation [11], human embryos show mesodermal structures already prior to the onset of the PS [4,26]. These cells reside in the space between the parietal endoderm (a derivative of PrE) and trophoblast [4,27]. The ExM cells delaminate from the parietal endoderm, whereas the trophoblast layer is separated by a basal lamina [4,27]. Thus, the parietal endoderm, originating from PrE, represents the logical prospective source of human ExM [4,26,27].

In this study, we established a protocol to induce ExM-like cells from nEnd corresponding to PrE [28] and thus a developmentally sound source. Using bulk and scRNAseq and tube formation assay, we established that nEnd cells have the potential to differentiate towards ExM lineages. Ultimately, these findings provide guidance for generating human ExM-like cell culture that could contribute to innovative applications for cellular assays and organoid models.

## 2. Results

### 2.1. Evolutionary and Molecular Characterization of Human ExM

The mechanism of human peri-gastrulation development, including ExM formation, remains understudied. While it is commonly acknowledged that mouse ExM is derived from the PS during gastrulation, human ExM is known to preexist before gastrulation takes place [4,11,26]. Thus, we reviewed published reports describing peri-gastrulation embryos of various eutherian species to gain insight into the evolution of the ExM derivation and to further substantiate the hypothesized human PrE origin (Figure 1A). The appearance of mesenchymal cells, resembling ExM, prior to PS formation has been reported in many species, including cattle [29], sheep [30,31], pig [32], horse [33] and armadillo [34]. Among Glires, ExM emerges through PS in murids (mouse [11] and rat [35]), whereas mesenchymal cells appear already in the pre-PS rabbit embryo [36]. These findings suggest that murids are an anomaly among eutherians concerning ExM formation. This aspect could be associated with the cylindrical shape of the murine gastrula [11,35]. The human peri-gastrulation epiblast is a disc [4], similar to other eutherians. The embryonic morphology and the timing of ExM formation suggest that human ExM originates from a non-PS source, and the PrE represents a plausible source (Figure 1A).

Considering these evolutionary aspects, we hypothesized that human ExM cells could be derived in vitro from nEnd [28]. Epithelial cells undergo epithelial-to-mesenchymal transition (EMT) in response to the BMP4 signal [37]. ExM cells delaminating from the parietal endoderm morphologically resemble EMT-like cells [27]. Thus, we applied the human mesoderm induction protocol with high BMP4 levels [38] on nEnd to induce EMT and ExM-like characteristics. Interestingly, this protocol promotes ExM derivation from mouse epiblast stem cells [39], while inducing lateral plate mesoderm (LPM) from human embryonic stem cells (ESC) (Figure 1B).

To identify suitable markers for human ExM, we analyzed scRNAseq data from an in utero human gastrulating embryo (Figure 1C) [19]. To accurately discern the ExM cluster, we identified a combination of positive and negative markers, *LUM*^pos^/*ANXA1*^pos^/*LIX1*^neg^. LUM is an extracellular matrix (ECM) protein found in several tissues, including the placenta [40], while ANXA1 is associated with the ECM and has a wide range of functions including endothelial regulation [41,42]. The expression of *Lix1* has been reported in later-stage chick embryonic tissues, such as in the nascent limb buds and in the pharyngeal and foregut mesenchyme [43]. For its absence from the extraembryonic tissues, *LIX1* has been chosen as the negative marker. In addition to these markers, we considered that the PS marker *BRACHYURY* (*TBXT*) should be negative throughout differentiation, as human ExM does not emerge from the PS [4,27].

**Figure 1 ijms-24-11366-f001:**
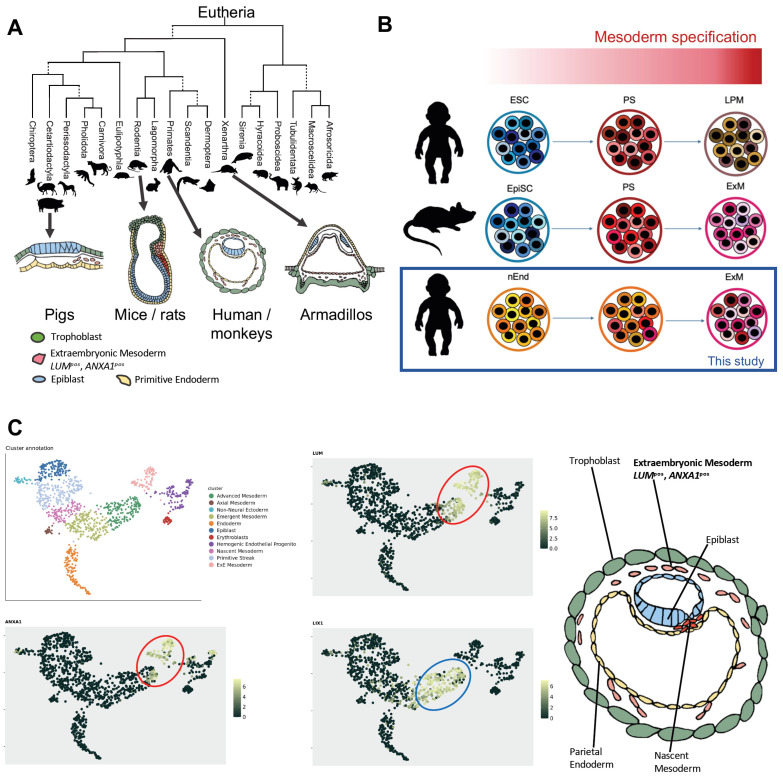
Comparison of morphology of eutherian embryos at peri-gastrulation stage. (**A**) Evolutionary tree representing the morphological differences among eutherian embryos. Pig embryos, belonging to the Cetartiodactyla, show a disc-like shape. The mouse and rat embryos, belonging to the Muridae family under the Rodentia order, have a unique cylindrical shape, while the armadillo embryo’s epiblastic plate has a cup-like structure. Primate embryos such as humans and monkeys instead resemble an oval-shaped disc. ExM formation precedes the establishment of the PS in all shown species except of the mice and rats. The proximity makes the PrE the prospective source of ExM. Green, trophoblast; blue, epiblast; pink, ExM; yellow, PrE and its derivatives; red, PS. (Phylogenetic tree was adapted from [44]). (**B**) Illustration summarizing the derivatives obtained by treating the in vitro pluripotent cells with a mesoderm induction protocol with high BMP4 levels. Primed ESCs induce PS and generate LPM. Mouse epiblast stem cells and human nEnd generate ExM instead, which is the subject of this study. (**C**) UMAP visualization of scRNAseq data from a gastrulating human embryo showing the cluster annotation of the different cell types, the expression of *LUM* and *ANXA1* (red circle) and the expression of *LIX1* in the embryonic mesoderm (blue circle) (http://www.human-gastrula.net/ [19], accessed on 19 December 2022). Right panel, schematic representation of the human gastrula showing the cells delaminating from the parietal endoderm that generates the ExM marked by *LUM* and *ANXA1* expression.

### 2.2. ExM Genes Can Be Efficiently Induced from nEnd Cells by Mesoderm Induction Factors

To assess if ExM can be derived from nEnd, we applied the above-mentioned mesoderm induction protocol on nEnd cells and analyzed the gene expression profiles on day 0, 1, 2, 8, and 15 by RT-qPCR. We also subjected naïve ESCs (nESC) and primed ESCs (prESC), representing pre- and post-implantation embryonic epiblast cells, respectively [45], to the same mesoderm induction protocol (Figure 2A). As expected, nEnd cultures showed no *OCT4* expression, a marker for the embryonic pluripotency (Figure 2B) [28,46,47]. Importantly, the PS marker *TBXT* was not expressed in nEnd cultures at any time point, whereas it peaked in prESC cultures on day 1, corresponding to the PS state. A weaker *TBXT* peak was detected in day 1 nESC cultures, suggesting that the nESC cultures contained a small fraction of cells that can directly differentiate towards PS (Figure 2B). Notably, *LUM* expression was significantly increased on day 15 in nEnd cultures compared to prESC and nESC, indicating that nEnd cells have a higher ExM conversion efficiency (Figure 2C). The negative marker *LIX1* exhibited low expression levels in all conditions on day 15 (Figure 2D). Furthermore, late-stage nESC cultures showed enrichment of the amnion marker *TFAP2B* [48] as well as the trophoblast marker *VGLL1* (Figure 2E) [49]. Here, *ISL1*, also associated with amnion [50], exhibited an increasing expression trend (Figure 2E).

These results imply that nESC have the potential to form both trophoblast and amnion cells in response to the mesoderm induction cocktail but not ExM-like cells. These data are also in agreement with the recently reported shared transcriptional features between trophoblast and amnion specification [25,51]. Day 15 prESC, on the other hand, contained beating cells resembling cardiac mesoderm (Appendix A), indicating that the high-BMP4 condition induces embryonic mesoderm from prESC. Thus, using the mesoderm induction protocol, ExM markers could most efficiently be induced from nEnd cells among the tested cell types.

### 2.3. Bulk RNAseq Reveals Gene Expression Pattern Corresponding to Gradual ExM Specification

Next, we employed bulk RNAseq to assess ExM formation over time (Figure 3A,B). To this end, RNA samples from differentiating nEnd cultures on day 0, 1, 2, 8, and 15 were subjected to bulk RNAseq. Principal component analysis (PCA) revealed distinct transcriptomics for samples collected at different time points (Figure 3C). We observed a decrease in *LIX1* expression levels and an increase in ExM markers *ANXA1* and *LUM* along the differentiation trajectory (Figure 3D). Additionally, we observed an increase in the expression of multiple genes associated with EMT, ECM, and blood and vasculature formation throughout the experimental timeframe. This expression pattern indicates a potential differentiation towards hemoendothelial lineages. Notably, *ISM1*, *SNAI2*, and *SPOCK1* genes associated with EMT [52,53,54,55,56] were enriched on day 0. Genes upregulated on day 1 included *MSN*, *NR2F*, and *PDPN*, also implicated in EMT [57,58,59,60]. At this stage, an increase in EMT and cell migration signature genes is expected since ExM cells delaminate from the parietal endoderm [27]. On day 2, many enriched genes were associated with ECM, including *ADAM12*, *COL5A1* [61,62], and *POSTN*, an amnion marker enriched also in ExM [19,63]. Gene expression patterns on day 8 and 15 were similar, and upregulated genes included *KDR* required for vasculogenesis and hematopoiesis [64], *EMP2* implicated in placental vasculogenesis [65], *DPP4* encoding a surface protein that controls the hematopoietic system and angiogenesis [66,67], and *PARM1* encoding a less characterized surface protein (Figure 3D).

In line with this observation, gene ontology (GO) term analysis of enriched genes on day 15 compared to day 0 underscored biological processes associated with tube formation and cell mobility (Figure 3E). Taken together, these results suggest that the mesoderm induction cocktail promotes gradual ExM-like specification from nEnd cells. Furthermore, we identified PARM1 and DPP4 as potential novel surface markers for human ExM. To avoid ambiguity, we will call day 15 nEnd culture PrE-derived ExM-like cells (PrxM) hereafter to distinguish it from the in vivo ExM lineage.

### 2.4. Single-Cell RNA Sequencing Reveals Distinct Cell Populations within PrxM

To gain insight into the composition of the PrxM cell population, scRNAseq was performed using the 10x technology from cryopreserved cells. A total of 944 cells that passed the quality control were used for further analysis. Uniform Manifold Approximation and Projection (UMAP) clustering revealed a predominantly homogeneous cell population that could be divided into three subclusters with distinct gene expression profiles (Figure 4A,B). The key ExM markers *LUM* and *ANXA1* were highly and largely homogeneously expressed in all subclusters, and the negative marker *LIX1* levels were generally low (Figure 4C–E). The putative ExM marker *PARM1* found by bulk RNAseq was also expressed in all subclusters with an enrichment in subcluster 2 (*p*adj = 2.66 × 10^−13^) (Figure 4F). On the contrary, another presumed ExM marker found by bulk RNAseq, *DPP4*, was increased in subcluster 3 (*p*adj = 4.97 × 10^−11^) (Figure 4G).

We further performed GO term analysis to address the relevant pathways in each subcluster. Subcluster 1 was characterized by an abundance of oxidative phosphorylation and ribosome (both *p* < 2.2 × 10^−16^) genes, suggesting an overall distinct metabolic state that demands higher energy [68,69,70]. These cells could represent a less differentiated population with a higher cell growth or proliferation index. Among the elevated pathways in subcluster 2 were the gonadotropin-releasing hormone (GnRH) signaling pathway (*p* = 1.36 × 10^−5^), TGF-β signaling pathway (*p* = 3.84 × 10^−5^), parathyroid hormone (PTH) synthesis, secretion and action (*p* = 4.31 × 10^−5^), and relaxin signaling pathway (*p* = 9.08 × 10^−6^). GnRH, TGF-β, PTH, and relaxin are all produced in the placenta, where they have pivotal roles in ensuring the integrity of the maternal–fetal interface and embryonic development [71,72,73,74,75,76,77,78]. Thus, this population could represent the placental ExM cells supporting pregnancy by hormone and growth factor signaling pathways. Subcluster 3 was enriched in adherens junction (*p* = 1.52 × 10^−9^) and axon guidance (*p* = 1.11 × 10^−8^) genes, suggesting that these cells are characterized by controlled cell mobility and possibly play a role in vascular integrity and invasion [79,80].

In conclusion, we identified three subclusters within the PrxM population that revealed distinct putative roles in growth, hormone signaling and endothelial integrity. However, due to missing data from in vivo counterparts at this specific developmental stage and resolution, it is challenging to attribute a specific cell type and function to these subclusters.

### 2.5. PrxM Shares Cell Populations with Gastrulating Human Embryo

To address the suitability of PrxM cells as an in vitro model for human ExM, we evaluated if and to what extent the PrxM population corresponded to ExM in the gastrulating embryo. To this end, scRNAseq data from our study were integrated with the data from the gastrulating human embryo previously used to determine the ExM markers [19]. UMAP clustering showed that *LUM* and *ANXA1*-expressing populations from the gastrula overlapped with PrxM, indicating comparable cell identities with ExM (Figure 5A). However, the overlap was only partial, possibly due to the presence of populations that might not be represented in the gastrula at this stage. Nevertheless, the PrxM population contains cells closely resembling ExM of human gastrulating embryo.

To gain a deeper understanding of PrxM and assess its suitability as an ExM model, we conducted a comparative analysis between the cell composition of PrxM and existing in vitro ExM-like cells [81]. In this model, naïve pluripotent stem cells were treated with the trophoblast stem cell medium called ASECRiAV, in which the mesenchymal CDH1^neg^ population, showing ExM-like gene expression profiles, were cultured for up to 70 days [81]. We integrated the PrxM data set with day 70 ASECRiAV cells (ASECRiAV) [81]. The clustering analysis revealed minimal overlap between the two populations, despite comparable expression of ExM markers in both groups (Figure 5B). Therefore, both PrxM and ASECRiAV could potentially represent distinct ExM-like populations.

We proceeded to compare PrxM cells with the previously published scRNAseq data obtained from in vitro PrE and nEnd cells [28,81]. Although these populations serve as the source for PrxM, previous studies have shown that they cluster together with ASECRiAV cells [81]. Consequently, we aimed to determine their similarities and differences with PrxM cells. Our analysis revealed minimal overlap between the PrE, nEnd, and PrxM populations, with higher expression levels of *LUM*, *PARM1*, and *DPP4* observed in the PrxM fraction compared to PrE and nEnd (Figure 5C). Collectively, these findings suggest that PrxM represents a distinct cell population that closely resembles ExM cells present in human gastrulating embryos.

### 2.6. PrxM Shows Angiogenesis Capacity on Soft Matrix

To determine whether the increased expression of angiogenesis-associated genes and the resemblance to in vivo ExM population confer functional properties to PrxM, we evaluated the angiogenesis potential of nEnd cells undergoing differentiation towards an ExM-like state on days 0, 1, 2, 8, and 15 (Figure 6A). We employed the tube formation assay, which is widely used to determine either the vessel formation capacity of cells or (anti-)angiogenic effects of compounds in the media [82,83]. Since the basal medium used in this study, N2B27, contains the pro-angiogenic factor insulin [84,85], we seeded cells at the above-mentioned time points on growth factor-reduced soft matrix and in basal N2B27 medium and incubated them for 1 day (Figure 6A,B). The number of tubes, nodes, and the tube/node ratio were quantified to characterize the vascularization pattern. Already on day 0, nEnd cells exhibited vascularization potential, which increased on day 1 but drastically decreased on day 2. Subsequently, the vasculogenesis activity remained relatively low until day 8, but unexpectedly, it showed a partial restoration on day 15 (Figure 6C), despite the similar gene expression patterns observed between day 8 and day 15.

These results highlight the existence of two distinct windows of vasculogenesis competency in vitro on day 0–1 and, later, on day 15. The cells on day 0–1 exhibited higher levels of EMT markers which is often linked to the ability of cells to form vessels in tumors [86,87], suggesting that the initial window of tube formation is associated with EMT and delamination. Conversely, the later window could correspond to the endothelial vasculogenic properties associated with ExM. Altogether, these findings indicate that PrxM cells harbour vasculature-forming properties.

## 3. Discussion

The first weeks of human embryonic development are of critical importance, although extremely challenging to address, due to the ethical issues surrounding the use of human embryos for research purposes [12,13]. To overcome this difficulty, recent studies have implemented cell line-based models of human embryogenesis resembling implantation, symmetry breaking, and gastrulation [15,20,21,22,24,88]. In this regard, human ExM is still an elusive cell population, with limited information concerning its origin and in vitro representation. While existing evidence suggests that in vivo ExM does not originate from the epiblast [18,26], the only currently available protocols to generate ExM-like cells in vitro require the cells to traverse the epiblast state [81,89].

In this study, after reviewing peri-gastrulation development in mammals, we established a novel approach based on the hypothesis that ExM is derived from PrE. We explored the in vitro differentiation potential of nEnd, which is a PrE cell line. Using a mesoderm induction protocol, we differentiated nEnd towards a cell population with an ExM-like gene expression signature and termed this population PrE-derived ExM-like cells, PrxM. We also exposed the pluripotent prESC and nESC populations representing the cell lineages existing in the embryo around the same time as PrE to the same signaling cues. While prESC and nESC generated embryonic mesoderm and a mixture of trophoblast and amnion-like cells, the induction of ExM markers was most effective from nEnd cells, highlighting their excellent potential as a source for in vitro ExM-like cells. During the differentiation, cells sequentially expressed the markers for EMT, ECM, and blood and vasculature formation, resembling the physiological development of ExM, and recapitulating delamination from the parietal endoderm layer, yolk sac formation, hematopoiesis, and vasculogenesis. Single-cell RNAseq analysis of PrxM cells showed homogenous expression of known ExM markers within the population. By integrating and performing UMAP clustering on our dataset and published data, we discovered that a portion of PrxM cells overlapped with cells found in the human gastrulating embryo. This finding strongly suggests that PrxM successfully mimics the ExM population. Furthermore, we observed that the PrxM population displayed notable potential for vascularization in vitro when cultured on a soft matrix. Vasculogenesis is a critical aspect of the ExM function in the developing embryo.

The origin of human ExM is a highly debated topic. Histological samples of human and rhesus monkey embryos suggest that ExM emerges from PrE [26,27]. This hypothesis has been further substantiated by a recent computational study examining DNA mutations in embryos [18]. While the epiblast origin of ExM cannot be excluded, our study shows that nEnd cells generate ExM-like cells, PrxM, exhibiting similar expression as the ASECRiAV cells. These findings indicate that distinct ExM-like cell types can be generated from different sources and agree with a proposed but not yet experimentally supported hypothesis that PrE is the initial source of the ExM population, which is later completed by ExM cells of epiblast origin [4].

Tissues derived from multiple cell sources are a common occurrence during embryonic development. One notable example is the nervous system, which comprises neurons in the brain originating from the neuroectoderm and the spinal cord derived from neuromesodermal progenitors that pass through the PS, indicating a mesodermal origin [90]. Therefore, it is tempting to think that the situation is comparable with the sources of ExM, with cells originating from different lineages having the ability to contribute to the development of the same tissue.

Another possible source for human ExM that can be speculated based on the morphological proximity is the amniotic membrane. This layer is derived from epiblast and shares common markers with ExM [4,81,88]. Therefore, this perspective aligns with the concept of dual or multiple cell sources contributing to ExM development. On the contrary, the trophoblast origin of ExM is considered unlikely due to the presence of a basal lamina that separates the ExM and the trophoblast compartments [27]. However, trophoblast, mesoderm, and amnion share several markers, some of which are associated with BMP4 signaling response [20,25,51,81,89,91,92], implying that these cell types might have common regulatory networks that could allow them to interconvert in certain culturing conditions. Thus, it might be feasible to derive ExM from trophoblast stem cells in vitro.

However, it is important to note a general limitation of in vitro studies regarding the derivation and segregation of lineages, as the observations may not fully represent in vivo development. In this context, we often rely on evidence from rare ex utero embryo samples, although the reliability of such evidence is compromised due to the limited number of available samples.

In this study, we aimed to develop a chemically defined formula to minimize uncontrolled fluctuations and establish a reliable and consistent protocol. In contrast, the generation of ExM using ASECRiAV medium [81] involves the use of BSA and FBS, which are not chemically fully defined, and also requires the sorting of CDH1^neg^ ExM cells. These factors make it challenging to dissect the specific signaling pathways involved in driving ExM differentiation. However, through a comparison of the protocols for PrxM and ASECRiAV, we can identify potential signaling cues that contribute to ExM induction. One common factor between the protocols is the use of A8301, an inhibitor of the ACTIVIN/NODAL/TGF-β pathway. The inhibition of this pathway has been associated with the regulation of endothelial development [93] and hemangioblast formation [94]. In line with this observation, TGF-β1 has been reported as the inhibitor of hematopoietic stem cell proliferation [95,96]. Thus, A8301 might account for the induction of endothelial and hematopoietic properties in ExM. Another common component is insulin, which could promote ExM induction via PI3K/AKT/mTOR pathway, accounting also for its pro-angiogenic role [81,84,85].

It is important to highlight that there are notable differences between the two protocols. The ASECRiAV medium includes EGF but lacks BMP4, which is in contrast to our protocol. The significance of BMP4 in proper PS and mesoderm development has been well documented [20,37], and it is known to be secreted from the amnion during mesoderm induction [50]. However, BMP4 and EGF could be redundant during ExM formation, possibly via the common PI3K/AKT and/or MAPK pathway [97]. Additionally, the two protocols also contain other components with opposing effects. Our protocol involves a long-term treatment with WNT inhibitor C59. In contrast, the ASECRiAV medium contains WNT agonist CHIR99021, implying a differential role of the WNT pathway in PrE- and epiblast-derived ExM formation.

Nevertheless, our data imply room for further protocol optimization, since the in vitro differentiation of nESC towards PrxM takes longer than 2–3 weeks, as opposed to the normal ExM development time in vivo [19]. One potential approach to expedite ExM specification could be the addition of growth factors such as EGF, which may have synergistic effects with BMP4. Nonetheless, it is crucial to perform experiments where individual components of the medium are removed to identify the essential factors to refine the protocol. Studying ExM can critically impact our understanding of human development and disease modeling for several reasons. Abnormalities in ExM development have been linked to various pregnancy-related complications, such as preeclampsia, placental insufficiency, and fetal growth restriction.

Thus, understanding the development and function of ExM can shed light on the intricate processes that sustain embryonic growth and ensure healthy embryonic formation, requiring a cellular model to study the mechanisms of ExM disorders and develop strategies for their early detection, prevention, and treatment.

In conclusion, this study has successfully demonstrated the ability of nEnd cells to differentiate towards an ExM-like population, PrxM, characterized by the expression of lineage markers and angiogenic potential. Thus, PrxM cells provide a valuable resource for advancing research in early human development and associated diseases.

## 4. Materials and Methods

### 4.1. Cell Culture

H9 cells (WA09, WiCell, Madison, WI, USA) were used for all experiments. Naïve ESCs were maintained as described [98] in t2iLGö/tt2iLGö medium on irradiated MEF feeders. Derivation of nEnd cells from nESC has been described elsewhere [28]. nEnd cells were maintained under feeder-free conditions on 10 μg/mL Biolaminin 521 (BioLamina, Sundbyberg, Sweden, LN521) in N2B27 medium [99] supplemented with 10 ng/mL LIF (made in house), 3 μg/mL CHIR99021 (Axon MedChem, Groningen, The Netherlands, 1386), 100 ng/mL ACTIVIN A (PeproTech, Rocky Hill, NJ, USA, 120-14E), and 1:1000 Penicillin–Streptomycin (Thermo Fisher Scientific, Waltham, MA, USA, 15070063), and routinely passaged with Accutase (Thermo Fisher Scientific, A1110501) in medium containing 10 μM ROCK inhibitor (VWR, Radnor, PA, USA, 688000) for the first 24 h after passage. Both nESC and nEnd were cultured under hypoxic conditions (5% O_2_, 5% CO_2_ at 37 °C). Primed ESCs were maintained in mTeSR1 (STEMCELL Technologies, 85850) on 5 μg/mL Biolaminin 521 under normoxic conditions (20% O_2_, 5% CO_2_ at 37 °C) and routinely passaged with TrypLE Select (Thermo Fisher Scientific, 12563011) without using ROCK inhibitor. All cell types were regularly screened for mycoplasma and tested negative.

### 4.2. Cell Differentiation

The ExM induction protocol was modified based on previous reports [38,39]. Prior to induction, all cell types were brought into the same hypoxic, feeder-free condition on 10 μg/mL Biolaminin 521. For this, nESC were depleted of feeder cells [24] and prESC were placed in the hypoxic incubator 24 h after plating. After reaching confluency, cells were briefly washed with PBS and incubated in N2B27 medium supplemented with 30 ng/mL ACTIVIN A, 40 ng/mL BMP4 (PeproTech, 120-05ET), 6 μM CHIR99021, 20 ng/mL FGF2 (PeproTech, 100-18B), 100 nM PIK90 (Axon MedChem, 1362), and 1:1000 Penicillin–Streptomycin. After 24 h, following a brief wash with PBS, cells were treated with N2B27 medium supplemented with 1 μM A8301 (Sigma, St. Louis, MO, USA, SML0788), 30 ng/mL BMP4, 1 μM C59 (Axon MedChem, 2287), and 1:1000 Penicillin–Streptomycin for 14 days, with a medium change of every 1–2 days.

### 4.3. RNA Extraction

RNeasy kits (Qiagen, Hilden, Germany, 74104 and 74004) were used to extract RNA. On day 0, 1, 2, 8, and 15, cells were briefly washed in PBS and collected in RLT buffer with a scraper. All following procedures were performed according to manufacturer’s instructions.

### 4.4. RT-qPCR

Reverse transcription was performed using Random Hexamers (Thermo Fisher Scientific, N8080127) for annealing and SuperScript III Reverse Transcriptase (Thermo Fisher Scientific, 18080044) for cDNA synthesis. Quantitative PCR reactions were executed on LightCycler 480 II (Roche Diagnostics, Basel, Switzerland) using SYBR Green (Roche Diagnostics, 04707516001). Standard curves were created to determine the efficiency of reactions. *ACTB* and *GAPDH* were used as the housekeeping genes to normalize the gene expression levels. All primers are listed in the Appendix A.

### 4.5. Live-Cell Imaging of Beating Mesoderm

Primed ESC-derived beating cells on differentiation day 15 were imaged with AF6000 (Leica Microsystems, Wetzlar, Germany) at 37 °C. A total of 10 images were taken per second and compiled to videos representing the real beating speed.

### 4.6. Bulk RNAseq

RNA concentration was measured with NanoDrop 2000 (Thermo Fisher Scientific), and 120 ng were used for the library preparation with NEBNext Ultra II RNA library Prep Kit (New England BioLabs, Ipswich, MA, USA, E7770). PCR amplification was performed for a total of 9 cycles, and AMPure XP (Beckman Coulter, Brea, CA, USA, A63881) was used in purification steps. Fragment Analyzer (Agilent Technologies, Santa Clara, CA, USA) was used to assess the library quality, and the concentration was determined using Qubit Fluorometer (Thermo Fisher Scientific). Pooled libraries were sequenced on NextSeq 500 Sequencer (Illumina, San Diego, CA, USA) at reNEW/CPR Genomics Platform (University of Copenhagen, Copenhagen, Denmark), where the raw data were processed in a standardized workflow. The count table was analyzed with R/Bioconductor (v. 3.16) [100]. Normalization and identification of differentially expressed genes were performed with the DESeq2 package (v. 1.34.0) [101].

### 4.7. Single-Cell RNAseq

On day 15 of differentiation, nEnd-derived cells were dissociated with TrypLE Select and frozen in KnockOut Serum Replacement (Thermo Fisher Scientific, 10828028) with 10% DMSO. Single-cell RNAseq was performed by Single Cell Discoveries (Utrecht, The Netherlands). Cells were thawed in DMEM/F-12 medium (Thermo Fisher Scientific, 21331020) with 10% KnockOut Serum Replacement and subjected to the company’s standard workflow for 10x Genomics. Libraries were prepared with 3′ v3.1 kit and sequenced on NovaSeq 6000 (Illumina), applying PE150 mode. Raw data were processed according to the company’s standardized workflow. The CellRanger (v. 7.0.1) pipeline [102] output, filtered for empty barcodes, were analyzed with R/Bioconductor using the Seurat package (v. 4.3.0) [103].

High-quality cells were filtered using the argument nFeature_RNA > 5000 and nFeature_RNA < 12,000 and percent.mt < 5. Normalization was performed by LogNormalize method with a scale factor of 10,000, and cell cycle genes were regressed out. Differentially expressed genes were determined by FindMarkers function with Wilcoxon Rank Sum test, min.pct = 0.25 and logfc.threshold = 0.25.

### 4.8. Angiogenesis Assay

Flat bottom 96-well plates were coated with Cultrex with reduced growth factor (R&D Systems, Minneapolis, MN, USA, 3433-005-01) and incubated at 37 °C for 30 min or longer for polymerization. nEnd-derived cells on day 0, 1, 2, 8, and 15 of differentiation were dissociated and singularized with TrypLE Select and plated at 125.000 cells/cm^2^ in N2B27 medium supplemented with 1:1000 Penicillin–Streptomycin only. After incubating 1 day at hypoxic conditions, images of cells were taken with DMC4500 (Leica Microsystems) at 5× magnification. Tubes and nodes were counted manually using ImageJ (v. 1.53t) Cell Counter tool.

### 4.9. GO Term Analyses

Web-based gene set analysis toolkit (WebGestalt, https://www.webgestalt.org/, accessed on 30 November 2022) interface [104] was employed for GO term analysis. To compare the gene expression patterns of nEnd and PrxM, the genes with normalized counts ≥ 1000 in PrxM; *p*adj ≤ 0.05 and log2FC ≥ 1 were pre-selected. Over-representation analysis was run with the gene ontology database in the biological process category. To determine the enriched pathways in each single-cell-sequenced clusters, over-representation analyses were performed using KEGG pathway database.

### 4.10. Statistical Analyses

*p*-Values for RT-qPCR and angiogenesis experiments were determined by Student’s 2-sided *t*-test using t.test() function in R (v. 4.2). For the bulk RNAseq and scRNAseq experiments, statistical tests were performed using the respective R packages. Significance levels were defined as follows: not significant (ns) *p* > 0.05, * *p* ≤ 0.05, ** *p* ≤ 0.01, *** *p* ≤ 0.001, **** *p* ≤ 0.0001.

## Figures and Tables

**Figure 2 ijms-24-11366-f002:**
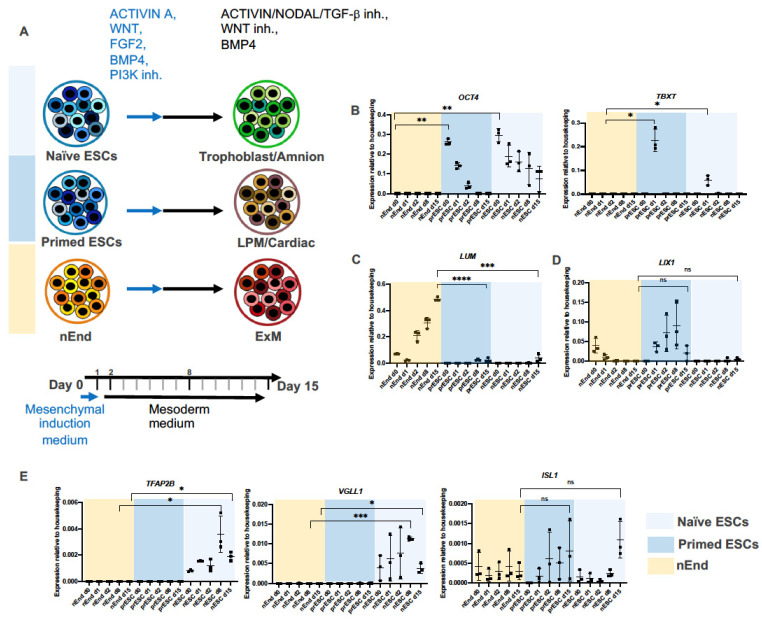
nEnd cell line is an optimal source for human ExM. (**A**) Illustration summarizing the effect of the mesoderm induction protocol on naïve and primed ESCs and nEnd. Cells are initially subjected to the mesenchymal induction medium for 1 day (blue arrow) and thereafter treated with the mesoderm medium for 14 days (black arrow). Specific lineage marker expressions tested by RT-qPCR indicate that naïve ESCs (light blue) differentiate toward trophoblast/amnion, prESC (dark blue) generate lateral plate/cardiac mesoderm, and nEnd (yellow) produce ExM-like cells. The time points selected for RT-qPCR are indicated in bold. (**B**) Relative mRNA expression levels of *OCT4* and *TBXT* measured by RT-qPCR, confirming high expression of the pluripotency marker in naïve and primed ESCs and PS induction in prESC. (**C**) RT-qPCR analysis indicates enrichment of the ExM marker *LUM* during differentiation of nEnd. (**D**) RT-qPCR reveals an increased trend of *LIX1* expression during prESC differentiation, which is characteristic of embryonic mesoderm. (**E**) Relative mRNA expression of *TFAP2B*, *VGLL1* and *ISL1* highlight the amnion/trophoblast-like profile of nESC derivatives on day 8 and 15. Error bars show means with standard deviations. Not significant (ns) *p* > 0.05, * *p* ≤ 0.05, ** *p* ≤ 0.01, *** *p* ≤ 0.001, **** *p* ≤ 0.0001 (*t*-test).

**Figure 3 ijms-24-11366-f003:**
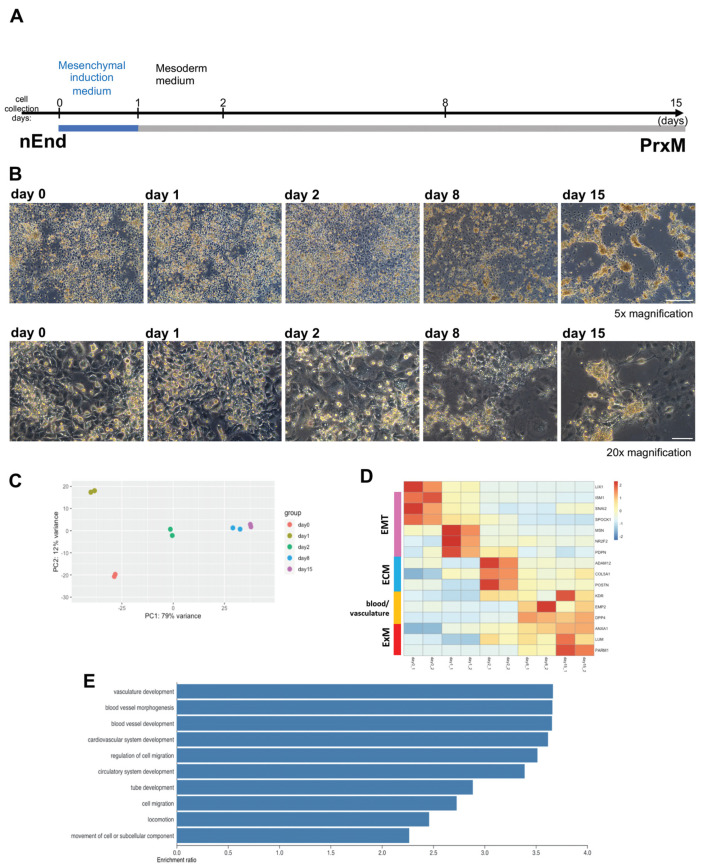
Mesoderm induction protocol converts nEnd cells into PrE-derived ExM (PrxM). (**A**) Schematic illustration of the sequential protocol used to differentiate nEnd into ExM-like cells termed PrxM. Cells are initially subjected to mesenchymal induction (blue) followed by 14 days of mesoderm differentiation (grey). (**B**) Representative brightfield images of nEnd differentiating towards PrxM, taken at different time points 5× magnification, scale bar: 500 μm; 20× magnification, scale bar: 100 μm. (**C**) PCA plot of bulk RNAseq results from different time points. While day 1 represents a unique EMT and cell delamination stage, day 8 and 15 cells have similar profiles. (**D**) Heatmap displaying relative expression levels of selected differentially regulated genes at different time points, showing upregulation of the ExM markers *LUM*, *ANXA1*, and *PARM1* along the differentiation trajectory. Pink, EMT; blue, ECM; yellow, blood and vasculature; red, ExM. (**E**) GO term enrichment analysis of genes differentially expressed in PrxM compared to nEnd indicates a significant overrepresentation of terms associated with vasculogenesis and cell migration.

**Figure 4 ijms-24-11366-f004:**
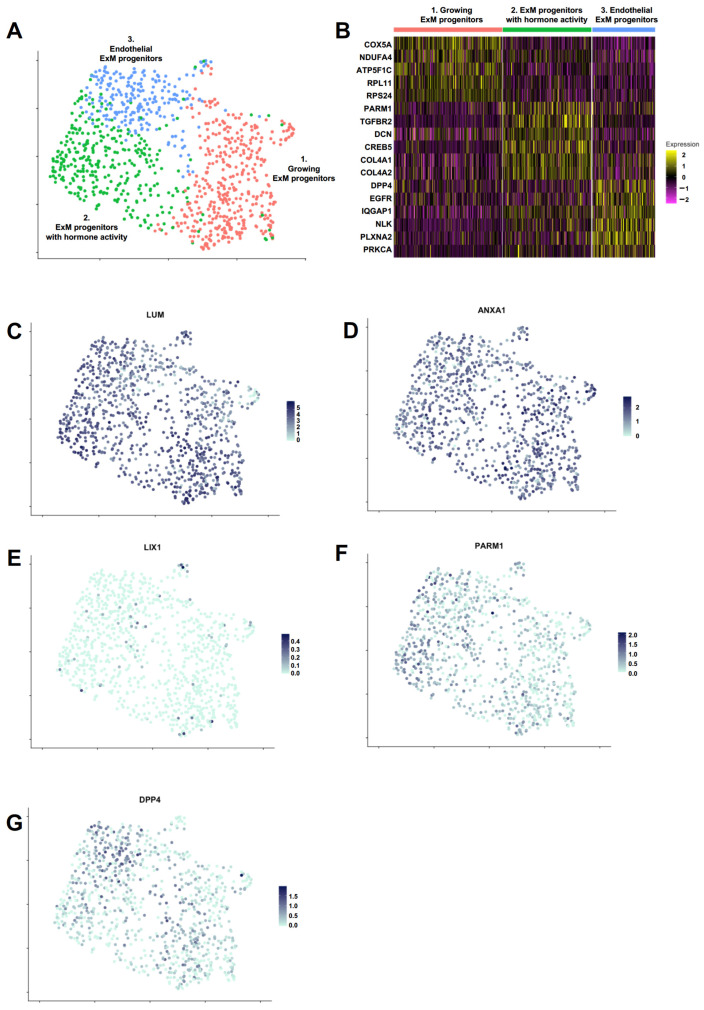
PrxM population includes three subclusters with distinct signatures in cell growth, hormone signaling and endothelial integrity. (**A**) Clustering of cells identified three subpopulations within PrxM. Cell identity was established by using significantly enriched genes in each subcluster. Growing ExM cells (red) express higher levels of oxidative phosphorylation and ribosome genes. ExM progenitors with hormone activity (green) exhibit higher levels of GnRH, TGF-β, PTH, and relaxin signaling pathway genes. Endothelial ExM progenitors (blue) are enriched in adherens junction and axon guidance genes. (**B**) Heatmap showing expression levels of selected marker genes in the three subclusters of PrxM. (**C**–**G**) UMAP plots of PrxM. Individual cells are coloured by the expression levels of key marker genes (*LUM*, *ANXA1*, *PARM1*, and *DPP4*). The embryonic mesoderm marker *LIX1* is absent in most cells.

**Figure 5 ijms-24-11366-f005:**
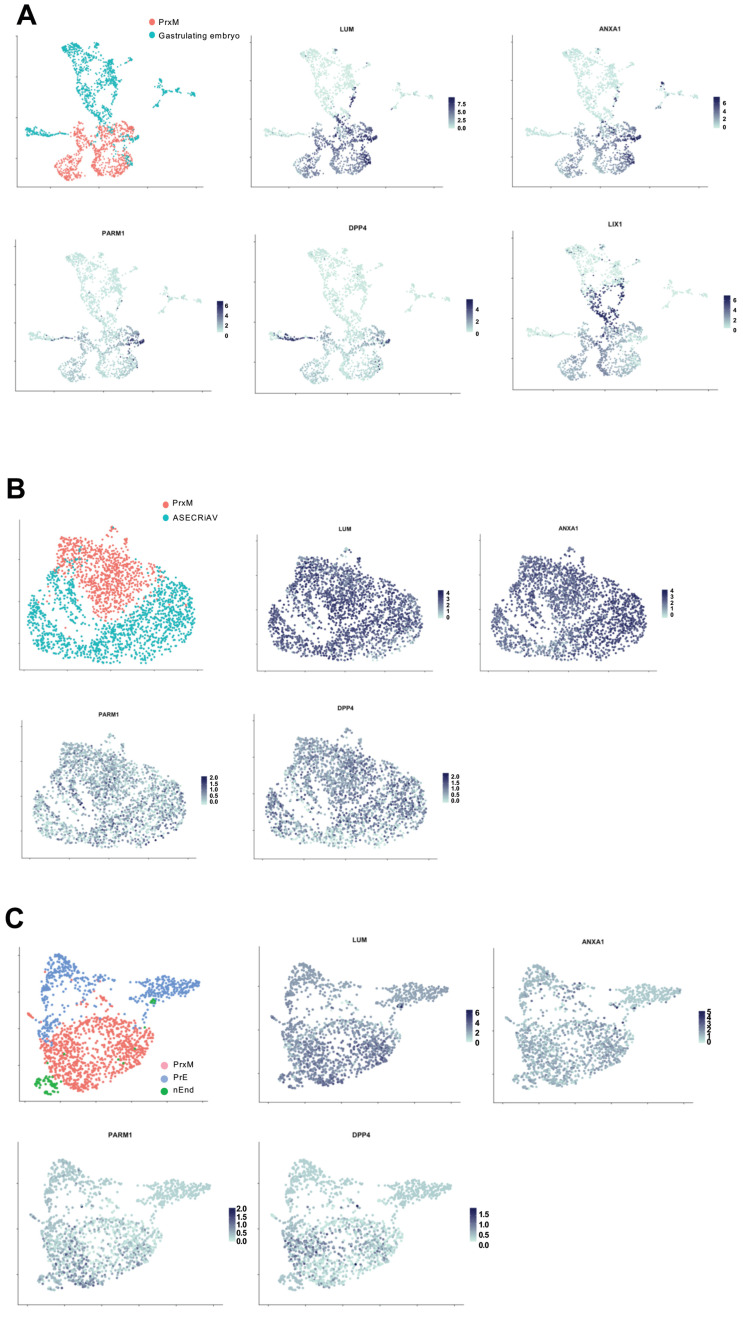
PrxM is a unique cell population that resembles ExM of human gastrula. (**A**) UMAP clustering of the integrated scRNAseq data from PrxM (red) and human gastrulating embryo (turquoise). Cell populations with relatively high levels of the ExM markers *LUM*, *ANXA1*, *PARM1*, and *DPP4* are shared between PrxM and the human gastrulating embryo, while embryonic marker *LIX1* is particularly enriched in the gastrula. The scRNAseq data of the human gastrula was obtained from http://www.human-gastrula.net/ [19] accessed on 19 December 2022. (**B**) Comparison of the scRNAseq data from PrxM (red) and ASECRiAV cells on day 70 (turquoise), which were generated by the exposure of nESC to the ASECRiAV medium [81]. There is almost no overlapping cell population upon integration of the two data sets. (**C**) Integration of PrxM (red), PrE (blue), and nEnd (green) scRNAseq data showing transcriptional profile segregation among these cell types. PrE and nEnd scRNAseq data are obtained from [81].

**Figure 6 ijms-24-11366-f006:**
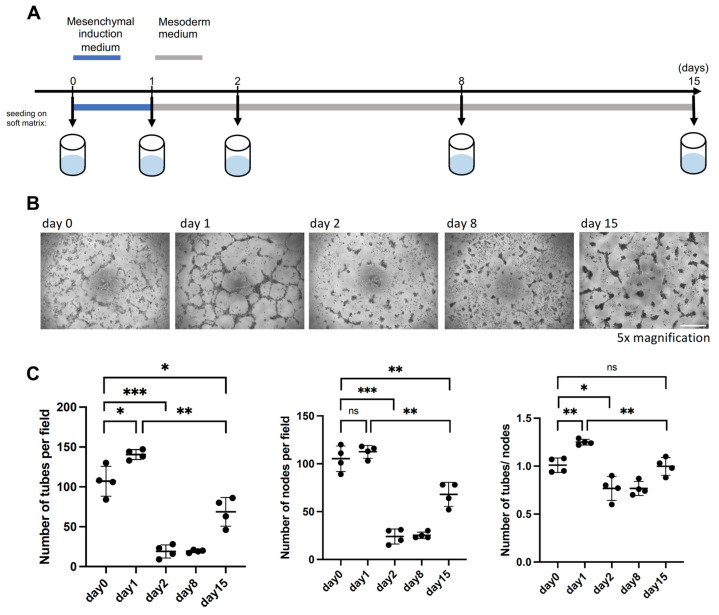
PrxM cells harbour vasculature-forming properties on soft matrix. (**A**) Schematic representation of the tube formation assay used to assess the angiogenesis capacity of nEnd cells differentiating towards PrxM cells. (**B**) Representative brightfield images of the tube formation capacity on day 0, 1, 2, 8, and 15. Scale bar: 500 µm (**C**) Quantification of the number of tubes, nodes, and the tube/node ratio to assess tube formation potential of nEnd cells along the differentiation trajectory towards PrxM. Error bars indicate means with standard deviations. Not significant (ns) *p* > 0.05, * *p* ≤ 0.05, ** *p* ≤ 0.01, *** *p* ≤ 0.001 (*t*-test).

## Data Availability

Bulk RNAseq and scRNAseq data from this study can be accessed from ArrayExpress under E-MTAB-12517 and E-MTAB-12562, respectively. Furthermore, the following published data were analyzed together with the data from this study: normalized gene expression matrix from http://www.human-gastrula.net/ [19] accessed on 19 December 2022 and GSE191286 [81].

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
