# Peer review of "Derivation of Human Extraembryonic Mesoderm-like Cells from Primitive Endoderm"

_ijms, 2023, doi:10.3390/ijms241411366_

Round 1

Reviewer 1 Report

The abstract is very complex

add scale bar to images

Figure 3. is not very clear

Please add scale bar Figure 6.

Poor, needs improvment

Author Response

We thank the reviewer for his/her comments and acknowledge their criticisms. We have rephrased the abstract to make it clearer. In the new version of the manuscript, we added the scaler bar in Figures 3 and 6. We also revised the entire manuscript to make it easier to read.

Reviewer 2 Report

Derivation of human extraembryonic mesoderm from primitive endoderm.

Farkas and Ferretti

This is an interesting paper that tackles the enigmatic extraembryonic mesoderm, of which its origin in human embryos is not understood. Indeed even in textbook images these cells are at times incorrectly  addressed or depicted. This manuscript will help to understand the origin and function of these cells. However, as it stands now, the conclusions drawn are too rigorous, and in my opinion it is not unequivocally demonstrated tat the cultured cells are indeed extraembryonic mesoderm. Mesoderm-like would be more appropriate.

In the title, it is suggested to change ‘extraembryonic mesoderm’ to ‘extraembryonic mesoderm-like cells’ as the cells that are formed are analysed by a small set of marker which is not enough to definitely identify these cells as mesoderm.

Lines 105-115, how do the ‘markers’ used distinguish extraembryonic mesoderm cells from extraembryonic (primitive and parietal) endoderm cells. In this respect immunohistochemistry in section of human or primate embryos. The use of LIX1 as a maker that should be absent from extraembryonic mesoderm seems not extremely discriminating. Have the authors considered markers that distinguish extraembryonic endoderm from extraembryonic mesoderm. LUM and ANXA do not seem able to fulfill this role (figure 1C).

With regard to angiogenesis and vasculogenesis, references describing that in the yolk sac and these events start from the extraembryonic mesoderm, rather that extraembryonic endoderm, are lacking. In other words besides the markers, how is it demonstrated that the cells are indeed extraembryonic mesoderm, rather than a more differentiated from of extraembryonic endoderm?  

Figure 1C, It is not clear that the schematic drawing of the human embryo indeed represents a gastrulating embryo as no embryonic mesoderm is present, not is there a primitive streak area visible in the epiblast.

Figure 2AB, what is the indication that the naïve EScs differentiated to trophoblast/ amnion? Differentiation to trophoblast would be remarkable as the cells have already passed the segregation time point from this cell type. For the amnion, how is amnion distinguished from epiblast in these cells? Expression of this gene in naïve ES cells (2E) indicates that TFAP2b is not an exclusive marker.

Primed ESCs in figure 2A are suggested to differentiate to LPM/ Cardiac, while in 2E it is demonstrated that these cells express ISL1 which according to the authors is a makers of amnion/ trophoblast. How is this explained.

Figure 2B what is the explanation for the specific expression of TBXT in prESCd1 and nESCd1 only (not other days)?

In figure 2 BCDE it seems that the statistical comparisons are used. For instance in E, It seems as if ISL1 expression in prESCday 15 is compared with nEnd day 15, but not with prESC d0? This should be properly explained.

Figure 5 and line 233-234 It is concluded that the LUM and ANXA1-espressing populations from the gastrula overlap with PrxM, indicating comparable cell identities with ExM. However, although cells in the gastrulating embryo can express these 2 genes, this does not mean that these cells are exclusively ExM. How can it be excluded that these cells do not (also) represent endoderm lineages?

Results presented in Figure5B suggest that the cultured cells are not extraembryonic mesoderm. Instead the authors conclude that PrxM cells represent a unique population resembling ExM. This is a bold conclusion. A better explanation for why these cells would be extraembryonic mesoderm and not another cells type (endoderm, or an in vitro-specific obtained cell type).

Minor

Refs 9, 13,  28, 29, 49, 70, 78 lack page numbers

Author Response

We thank the referee for the positive assessment of our manuscript and for acknowledging the novelty and importance of studying the origin and function of extraembryonic mesoderm. We also thank the referee for the suggestions that have helped us improve our manuscript.

As suggested by the reviewer, we toned down our claims of generating in vitro extraembryonic mesoderm (ExM) with our differentiation protocol. We agreed with the reviewer that we have a small set of markers to define extraembryonic mesoderm identity unequivocally. However, this is an unexplored field, and other bona fide markers are unavailable. We, therefore, termed ExM-like cells that we produced in vitro. Accordingly, the title is now: “Derivation of human extraembryonic mesoderm-like cells from primitive endoderm.” We have also substituted the term ExM with ExM-like when appropriate in the text.

We thank the reviewer for the comments and critical feedback about markers distinguishing extraembryonic mesoderm cells from extraembryonic (primitive and parietal) endoderm cells. The single-cell data presented in Fig. 1C, representing a gastrulating embryo, lacks annotated extraembryonic endoderm (PrE) or parietal endoderm (PE) cells. This is likely due to the limited number of cells with these specific identities. Consequently, the graph cannot be utilized to identify markers that distinguish ExM from PrE or PE. To the best of our knowledge, there is currently no available in vivo dataset that includes PrE, PE, and ExM cells, which would allow for direct gene expression comparisons among these cell types.

Although we acknowledge the reviewer's concern regarding the ambiguity of LIX1 expression in PrE or PE based on the graph, we cannot delve deeper into this matter due to the limited data availability. The presented dataset represents the only available information on human embryos during gastrulation. It is crucial to emphasize that obtaining such material at this early developmental stage is exceedingly rare and invaluable.

We thank the reviewer for raising the question about angiogenesis and vasculogenesis and the relationship between extraembryonic mesoderm and endoderm that in our original manuscript was unclear. The revised manuscript further clarifies that ExM is a more differentiated extraembryonic endoderm (PrE) form.

Based on the remarks of the reviewer, we have revisited Figure 1C, where we added a representation of the nascent mesoderm originating from the primitive streak.

We wholeheartedly concur with the reviewer's assessment of Figure 2A and 2B, which pertains to the differentiation of naïve ESCs into trophoblast and amnion lineages. It is indeed accurate to state that once cells have reached the trophoblast differentiation stage, they have already surpassed the segregation point. Nonetheless, it is worth noting that there have been reports of successful trophoblast induction from human ESCs, indicating that a subset of cells within the culture retains the potential to differentiate into trophoblast.

Here are a few examples to support this notion:

Horii et al., “ Human pluripotent stem cells as a model of trophoblast differentiation in both normal development and disease” Developmental Biology, 2016 https://doi.org/10.1073/pnas.1604747113

Li et al., “Establishment of human trophoblast stem cells from human induced pluripotent stem cell-derived cystic cells under micromesh culture” Stem Cell Research & Therapy, 2019 https://stemcellres.biomedcentral.com/articles/10.1186/s13287-019-1339-1

In respect to TFAP2B, we agree with the reviewer that TFAP2B is also expressed in naïve ESCs on day 0. However, it is elevated later during differentiation, together with ISL1 which is also an amnion marker. Moreover, in other studies, TFAP2A and GATA3 are listed as amnion markers but they label the trophoblast, too (Shao et al., “A pluripotent stem cell-based model for post-implantation human amniotic sac development”, Nature Communications, 2017, https://www.nature.com/articles/s41467-017-00236-w). Indeed TFAP2A and GATA3 are used as trophoblast markers in Krendel et al., “GATA2/3-TFAP2A/C transcription factor network couples human pluripotent stem cell differentiation to trophectoderm with repression of pluripotency” PNAS 2017. https://www.pnas.org/doi/full/10.1073/pnas.1708341114

Therefore, TFAP2B seemed to be a good amnion marker that is not expressed in trophoblast, as indicated also by Rostovskaya et al, (Amniogenesis occurs in two independent waves in primates”, Cell Stem Cell, 2022, doi: 10.1016/j.stem.2022.03.014)

We apologize for any confusion caused by our previous statement regarding ISL1 as an amnion marker. It is important to clarify that ISL1 is not solely specific to the amnion; rather, it is expressed in cardiac mesoderm as well. Given that a higher expression of ISL1 correlates with beating cells in prESC differentiation, it is more accurate to consider it as indicative of cardiac mesoderm rather than the amnion in this case.

Regarding Figure 2B, the specific expression of TBXT in prESCd1 and nESCd1 at day 1 indicates the induction of the primitive streak, which is a transient stage both in vitro and in vivo. After primitive streak induction, after day1, the cytokines present in the new media inhibit streak formation and promote the formation of an ExM-like population.

In Figure 2 BCDE, we focused on assessing the efficiency of differentiation, except in 2B where we examined the cell identity before differentiation (evaluated by OCT4 expression) and during differentiation (evaluated by TBXT expression). We provided statistical significance for the condition pairs that are particularly interesting and relevant to our study.

Regarding Figure 5, we acknowledge that the cell identity cannot be determined solely based on the expression of the ExM markers LUM and ANXA1. Therefore, we classify PrxM as ExM-like. However, referring back to Figure 1C, where it is evident that LUM and ANXA1 are not expressed in the endoderm (orange cell population), we can confidently reiterate that PrxM cells are ExM-like and do not represent the endoderm.

We update the references with page numbers. However, some of those are open-access articles and do not have page numbers.

We hope that, with all the changes listed above, the reviewer will find the manuscript more compelling.

Reviewer 3 Report

The author establish a new approach to set up ExM from PrE-like cells which has a higher differentiation potential. The article is well organized with a clear methodology, results and discussion. 

Very good. Minor typos only

Author Response

We thank the referee for expressing their enthusiasm for this work and appreciation.

Reviewer 4 Report

Congratulations to the authors for exploring their idea that ExM is derived from PrE. They have explored the in vitro differentiation potential of nEnd. They have developed a mesoderm induction protocol to differentiate nEnd towards a cell population with an ExM-like gene expression signature and termed this population PrxM. The concept and the execution is scientifically sound and I would recommend it for application without any further modifications.

Some minor language errors in the discussion that needs rectification.

Author Response

We thank the referee for the positive assessment of our manuscript and for acknowledging its novelty and importance.

Reviewer 5 Report

1. Please consider giving more context about the potential implications of your findings in the Introduction and Discussion sections. This could include implications for understanding human development, disease modeling, and perhaps regenerative medicine.

2. It would be helpful to include more discussion about the limitations of your current study and what future work is needed. For instance, while your in vitro model of ExM cells offers many advantages, how does it compare to actual human embryos in terms of gene expression and functionality? What aspects need to be improved upon?

3. The methodology is quite dense, and while this is somewhat unavoidable due to the nature of your work, consider if there are ways to make it more readable. Providing more detail about some of the techniques used (such as the specific steps involved in single-cell RNA sequencing) might also be helpful for replicability.

The English language quality of the manuscript is high, but the readability can be improved by simplifying the language, improving clarity, using consistent terminology, and conducting thorough proofreading.

Author Response

We thank the referee for his/her positive comments on our manuscript. We hopefully addressed all the concerns by discussing more clearly the data and novel insights that our protocol provides. In the revised manuscript, we discussed the implications and clinical perspective of our ExM differentiation protocol.

With respect to the methodology, we cannot provide more details about the single-cell sequencing methodology as it was done by the company and the company IP covers more details.

In our revised manuscript, we implemented the discussion and rewrote the manuscript to explain and discuss more clearly the data and novel insights our paper provides.

Round 2

Reviewer 1 Report

No more comments

No

Reviewer 2 Report

In general, the manuscript has been improved. Questions/ comments regarding the previous version have been well answered.

Minor issues:

Line 15 …from a PrE-like… instead of …from PrE-like…

As indicated in my previous report, references 9, 13,  28, 29, 49, 70, 78 lack page numbers. This has not been corrected.